# Optimization of the Cross-Sectional Geometry of Auxetic Dowels for Furniture Joints

**DOI:** 10.3390/ma16072838

**Published:** 2023-04-02

**Authors:** Tolga Kuşkun, Ali Kasal, Gökhan Çağlayan, Erkan Ceylan, Murat Bulca, Jerzy Smardzewski

**Affiliations:** 1Department of Woodworking Industrial Engineering, Faculty of Technology, Muğla Sıtkı Koçman University, Muğla 48000, Turkey; alikasal@mu.edu.tr (A.K.); gokhan.caglayan@yahoo.com (G.Ç.); 2Department of Furniture and Interior Design, Bingöl University, Bingöl 12000, Turkey; eceylan@bingol.edu.tr; 3Çilek Furniture Company, İnegöl 16420, Turkey; murat.bulca@cilek.com; 4Department of Furniture Design, Faculty of Wood Technology, Poznan University of Life Sciences, Wojska Polskiego 28, 60-637 Poznan, Poland; jsmardzewski@up.poznan.pl

**Keywords:** dowel optimization, numerical analysis, auxetic dowel, frame furniture, panel furniture, furniture joints

## Abstract

In this study, the aim was to optimize the cross-sectional geometry of auxetic dowels for furniture joints. For this purpose, two different sizes of auxetic dowels were chosen, one for frame- and the other for panel-type furniture joints for designing the cross-sectional geometry. Auxetic patterns that are created on the cross-sectional area cause deficiency of the materials, and this phenomenon decreases the modulus of elasticity (MOE) and increases the member stress. Accordingly, maximum MOE values and minimum Poisson’s ratio levels were determined for the optimum strength-auxetic behavior relation by means of a Monte Carlo method. Furthermore, Poisson’s ratio of the optimized dowel’s cross-section was confirmed with experimental tests, numerical analyses and analytical calculations. As a result, Poisson’s ratio values were obtained as negative values and confirmed, which means the dowels designed in this study had auxetic behavior. In conclusion, it could be said that studies should be conducted on the performance of auxetic dowels in both frame and panel furniture joints.

## 1. Introduction

In general, furniture is constructed by the frame (skeleton) or case (panel type) method, or by a combination of the two, known as the complex method [1]. Beam members are used to produce frame-type furniture, while panel members are used for case-type furniture. Chairs, tables and sofa frames are frame furniture; these are built with beam members and are also called frame construction. Case furniture is generally used for storage areas in kitchens, bedrooms, offices, etc.

The connection areas of the furniture are the most critical parts of furniture construction. Furniture connection can be created with different types of joints, such as glued or unglued joint types. Dowels, mortise and tenon joints are examples of the glued type of joints, and nails, screws, metal and plastic connectors are examples of the unglued joint type. Dowel joints are the most common joint type for the connection of furniture members. In the traditional type of dowel joints, wood and wood-based materials are connected to a wooden dowel by using glue [1]. Fasteners of ready to assemble (RTA) furniture joints become more popular with each passing day. This type of furniture joint is useful for shipping and transportation. Huge amounts of furniture volumes can be transported before assembly, and end-users can create the productions by means of RTA fasteners. Furthermore, shipping and assembly costs can be reduced by using RTA fasteners. This is a highly important issue for both domestic and exported furniture. When the topic is traditional furniture, it can be said that dowel joints have high strength values on the shear, torsional and bending forces. RTA furniture fasteners were created with an original design and operation. Besides new operation methods and design, new fasteners should carry the load which is applied to the furniture. In other words, RTA fasteners should carry tension, compression and bending loads as high as traditional furniture joints. New fasteners are externally invisible, quickly mounted and dismounted without tools. The finite element method (FEM) was used for the tests of this type of connections [2,3,4,5,6].

Using smart materials has become popular in many fields of industry. The design and production of smart materials comprises developing a new product or adding new properties to existing materials [7]. Auxetic behavior, which can be created to apply an auxetic pattern to the structures, is one of the fastest ways to obtain a smart material. The rule for obtaining auxetic behavior is related to the Poisson ratio, which should be negative on the auxetic smart materials. Materials have different behaviors under different types of loading, and Poisson’s ratio helps to define the behavior of the materials under loading. Poisson’s ratio can be calculated with strain of the transverse direction of the loading divided by to the strain of loading direction. Contrary to the positive Poisson’s ratio behaviors, auxetic materials become stretched longitudinally and become thicker under tension loading. Additionally, auxetic materials become shorter and thinner under compression loading. [8]. Structures which have a negative Poisson’s ratio were described almost 30 years ago [9,10,11]. They were made in 1987 by Lakes [12] and called auxetics by Evans [13]. The shear resistance synclastic behavior and indentation resistance of auxetic materials have better mechanical properties compared with traditional materials that have positive Poisson’s ratios. [14,15]. Using smart materials has been increasing in the industry. Studies of smart materials are increasing, but few reports deal with the implementation of auxetics in the furniture industry nowadays [16,17,18,19,20,21]. Auxetic patterns are commonly used in the research on materials. Different techniques can be used to create and produce auxetic materials [22].

The fundamental purpose of the mathematical models of auxetic materials is to predict the behavior of optimized geometry. While those mathematical models can theoretically work in any dimension, they have limited practical applications [23,24].

Limited numbers of studies handle the subject of auxetic materials and optimization techniques in the furniture industry. Those studies were established by Smardzewski et al. [16,25]. The aim was to produce a model of an auxetic spring for seating furniture constructions. Ren et al. [26] designed and produced the first auxetic nails for wood and the furniture industry. Load carrying capacities of auxetic and non-auxetic nails were compared under tension and compression loads. As a result, the performance of auxetic dowels was not at the same level as that of non-auxetic materials. Some suggestions were carried out about increasing the mechanical properties to designs of auxetic nails.

Kuşkun et al. and Kasal et al. subjected the auxetic dowels, which had auxetic patterns on the surface, under withdrawal and mounting force. Results showed that auxetic dowels could be alternative fasteners for traditional furniture dowels. According to the results, surface patterns cause a decrease in the mechanical performance because of the pattern’s cross-section. Concordantly, auxetic pattens could be applied to cross-sections of the dowels, and this product could be produced with both 3D printing technology and an injection molding system [1,7].

Optimization, in general terms, is all of the processes or methods that should be applied to make a system the most efficient at the least possible cost within certain constraints. In short, it is the process of optimally solving a problem or making a system the best possible. Optimization is one of the most basic requirements of engineering design. The mathematical definition of optimization, which is also a subject of mathematics, is to examine or solve a problem in a systematic way by selecting real or integer values in a certain and defined range and placing them into the function in order to minimize or maximize a real function. Optimization can also be called the art of making the right decision.

When optimizing a system, certain constraints, variables and implementation methods are involved. There are limitations in the optimization that must be performed for each system. There are also some variables for the optimization of the system. In general, the optimization steps are as follows:▪Analyzing the system;▪Identifying constraints;▪Identifying variables;▪Introducing the target (target function selection);▪Choosing the most appropriate optimization method;▪Control of the system.

Although studies in which optimization techniques are applied in various engineering fields are common in the literature, the number of studies in which these techniques are applied in furniture engineering is quite limited.

Optimization indicates ways to obtain the best results under determined conditions [27,28]. Optimization can be applied in every step of engineering studies, such as design, construction, operation and maintenance. One of the most commonly used types of optimization is structural optimization. “Structural optimization” indicates how to determine the best values or dimensions of structures under determined conditions, without the properties of the materials. Material is one of the most effective elements of engineering structures to obtain strong structures. Furthermore, composites are commonly used in every engineering field. The optimization of structures that include two or more types of materials is more difficult than that of structures that only include one type of material because of the computational difficulties. Structural optimization can be explained with the four sections below [28,29,30,31,32,33,34,35,36,37]:▪Size optimization: in other words, it can be described as sizing optimization, which deals with the cross-sectional areas of structures or cross-sectional areas of members of structures as the design variables;▪Shape optimization: Additionally, it can be described as configuration optimization, which deals with the nodal coordinates of structures as the design variables;▪Topology optimization: this optimization is the aim to delete needless structural members to reach the optimum design values;▪Multi-objective optimization: using two or more of the above optimization methods at the same time to produce better optimization results.

Tsiptsis et al. applied the Particle Swarm Optimization (PSO) method to investigate the shape and topology optimization problems of both 2D truss and frame towers. They indicated that if formulations can be prepared for every type of problem individually, solutions can be taken, and those methods can be used to solve structural and architectural problems [34].

Dapogny et al. prepared shape and topology optimization for conceptual architectural design. They collected personal opinions about the designs and formulated the optimization on the geometry of the shape. As a result, it was declared that three new functional shapes, which related to the mechanical performance of structures, were developed by means of personal aesthetic opinion [38].

The studies related to optimizing furniture systems are generally based on reducing the cross-sectional dimensions of the elements that make up the various types of furniture, and thus, the lightening of the system.

Smardzewski and Gavronski studied the optimum weight–strength relation by aiming to define the minimum material volume. According to experiments, material usage can be reduced by 53% of the original volume in the optimized chair by using the Monte Carlo method, which is combined into a FEM software [39]. It was reported that upholstered furniture in computer-aided engineering design obtained a method for optimizing the cross-sectional dimensions of the frame structure. As a result, consumption of beech reduced by 36% and particleboard by 25% [40]. Another result obtained from the study was that the optimized cross-section of the members was not negatively affected by the strength of the furniture frame [41]. Ke et al. produced L-shaped corner joints with dowels and optimized them by using FEM with the Taguchi method under compression test. Tenon diameter, tenon length and structure style were evaluated to determine the optimum factors and levels for the von mises stress using FEM. As a result of this study, the L-shaped corner joint in pine is 45° Bevel Butt in structure style, 24 in tenon length, 6 in tenon diameter and 20 mm in tenon gap [42]. The objective of another study was to create a lighter frame structure to optimize the volume of wooden material while the frame has the same loading condition. The FEM and MATLAB were utilized for the optimization of the frame. Results showed that the volume of the optimized frame was reduced by 58% compared with the non-optimized version [41]. Hu et al. determined the importance of the stretcher position of the chair on the mechanical properties by numerical and experimental analyses. In the results of the study, it was indicated that the load carrying capacity decreased first and increased with the rise of the height of stretcher positions. In addition, FEM results agreed with the experimental results by 10%. The relationship between loading capacity and stretcher position was generated by using the response surface method, and the correlation coefficient was 88% [43]. Güray et al. [44] optimized section sizes of chair members and indicated that the reduction was 32% of the total weight and volume for beech chairs, while it was 16% for pine chairs without sacrificing the performance required for domestic usage.

In this study, it was considered that the property of negative Poisson’s ratio could be used to design different types of dowels that have auxetic cross-sections for easier push-in and harder pull-out for the frame- and panel-type furniture joints. The designed, optimized and produced dowels could be utilized for one-time ready to assemble (RTA) frame- and panel-type furniture constructions. For this purpose, one auxetic pattern which applied the cross-section was chosen to obtain negative Poisson’s ratio and two different sizes of auxetic dowels (one for frame-, the other for panel-type furniture) were created.

An optimum strength–auxetic properties target was chosen while designing auxetic fasteners. It does not seem possible to keep the strength and auxetic properties at the highest level at the same time, since the strength of the material will be weakened while it is being imparted to the material. It is important in terms of strength that the loss in the cross-sectional area is at the lowest level while gaining auxetic properties. In order for the strength of the dowels to be high, the values with the highest modulus of elasticity were found at the values with the lowest auxetic property; in other words, it has been determined in previous studies that as the auxetic properties are increased, the modulus of elasticity will decrease due to the loss in the cross-section of the material. In this case, the optimum cross-sectional dimensions of the fasteners were determined by calculating the most suitable dimensions for the cross-sectional area while gaining auxetic properties.

Consumers and producers have different expectations in terms of economic benefits from industrial products. While consumers naturally expect to buy the best quality product at the cheapest price, manufacturers aim to produce the product at the lowest cost and provide maximum profit. Accordingly, it is important for manufacturers to minimize their production costs as well as to meet customer expectations. In this context, it is important for manufacturers to produce durable products with minimum cost. At this point, the Monte Carlo optimization technique was used to obtain maximum strength and minimum Poisson’s ratio for cross-sections of the designed auxetic dowels. This method provides flexible and extremely powerful techniques for solving many optimization problems, and is the invention, in some situations, whose behavior and outcome can be used to study. The effectiveness of numerical or simulated studies as a serious scientific pursuit is increased by the availability of modern computers [45].

Monte Carlo methods, like statistical resampling approaches in general [46,47], have lately come to be perceived as fundamental tools by which many formerly very difficult estimation problems become trivial [48]. A definition of a Monte Carlo method would be one that involves the deliberate use of random numbers in a calculation that has the structure of a stochastic process. It is referred to with a stochastic process in that it is a sequence of states whose evolution is determined by random events. In a computer, these are generated by a deterministic algorithm that generates a sequence of pseudorandom numbers that imitate the properties of truly random numbers [45].

Accordingly, two different sizes of auxetic dowels that can be produced by both a 3D printer and injection molding technology were designed and optimized by using the Monte Carlo method. Then, Poisson’s ratios of the dowels were compared with each other, which were obtained from numerical analyses, analytical calculations and actual test results.

## 2. Materials and Methods

### 2.1. Design of the Auxetic Dowels

Within the scope of this study, auxetic patterns were first determined to be adapted to dowel cross-section designs; after that, the developed auxetic patterns were converted into designs suitable for dowel cross-sections. From many auxetic patterns, one pattern was chosen to use in the geometry of dowel cross-sections in order to gain the auxetic feature. Example auxetic patterns for dowel cross-section structures are shown in Figure 1.

The auxetic pattern in Figure 1c was chosen to be optimized and applied to dowel cross-sections. This pattern can be easily applied as a honeycomb system and also singly, and the pattern has auxetic behavior in both applications. Furthermore, the 4-star pattern can be produced both with a 3D printing system and an injection molding system. Autocad Inventor software was used for modeling and also paid attention to ensure that the auxetic pattern structure had the most suitable technological parameters for dowel production in both 3D printer and injection molding methods. Designed and modeled auxetic dowels for frame- and panel-type furniture joints are shown in Figure 2a,b.

As seen in Figure 2a, the first type of dowel has a 50 mm length and 15 mm diameter; the second type has a 40 mm length and 10 mm diameter. Two different dimensions of dowels were chosen to be used in different furniture structures. These dimensions were decided by taking into account the material (wood or wood-based panel) thicknesses that are commonly used in the production of frame- or panel-type furniture. The first type of dowel, which has a 50 mm length and 15 mm diameter, was chosen for frame furniture (Figure 3a), and the second type of dowel, which has a 40 mm length and 10 mm diameter, was chosen for panel furniture (Figure 3b).

### 2.2. Optimization of the Cross-Sectional Geometry of Auxetic Dowels

A typical optimization problem is defined with objective function. The objective function and constraints are all nonlinear, giving rise to a typical nonlinear programming problem. A Monte Carlo method, which makes fundamental use of random samples in the excel visual basic, was used for optimization. For the solution, firstly, the objective function and constraints were defined, and then a master routine was written to call the objective function and constraints. After the calculations were performed, a solution was obtained.

In the cross-section of the designed auxetic dowels, optimization was carried out in order to determine the optimum dimensions that give the highest resistance and the lowest negative Poisson’s ratio.

An optimization algorithm was created with excel visual basic. Four parameters, including inner wall, auxetic pattern angle (*φ*), outer wall dimension and diameter of dowel, were named as *t_A_*, fiA (*φ*), *T_wd_* and *T_d_*, respectively (Figure 4a). Minimum and maximum levels of those parameters were chosen to find optimum levels. The calculated dimensions for the auxetic pattern of dowels by using algorithm were shown in Figure 4b.

Constraints related to the production technology and auxetic behavior criteria were effective factors in determining the minimum and maximum values. Minimum and maximum values of constraints are related with nozzle diameter for 3D printer technology and mold for injection molding systems. The angle of the pattern levels was chosen between 15° and 40° because the length of the arm of the star reached the negative values in the formulas. Because of the reasons above, minimum and maximum parameters were determined and given in Table 1.

Optimization algorithm was created and MOE and Poisson’s ratio of dowels calculated (Figure 5). *t_A_*, fiA, *T_wd_* and *T_d_* are parameters to constrain the optimization equation (Figure 4a). After determining the maximum and minimum limit of parameters, all dimensions needed for producing the optimum cross-section of a dowel were determined (Figure 4b).

All dimensions to create a cross-section of auxetic dowels signed with question marks at Figure 4b were calculated by using the algorithm in Figure 5. A quarter part of the auxetic pattern was subjected to optimization and calculation because the pattern used in this study was symmetric in both horizontal (x) and vertical (y) directions. Optimization was carried out for the analysis of some parameters of the dowel section, which is reflected in the details at Figure 6, in such a way as to show the highest auxetic property under strength conditions.

Considering the quarter of the dowel shown in Figure 6, half of the lateral force, *F*, is expected to act on point *A*. This force causes compression in the horizontal extension with thickness *t_B_*, so the first limit criterion is related to the compression of this part of the section (Equation (1)).
(1)$$\frac{\left|F\right|}{t_{A}}\leq \sigma_{comp,all},$$

Equation (1) is the allowable axial compression strength of the section parts. Another important part of the section is subjected to bending under the *F* load. In order to calculate the bending moment, it is necessary to determine the coordinates of the *A* and *B* points and, thus, to express the *r_A/B_* vector that can be written for the moment arm. The coordinates of point *A* can be represented as in Equations (2) and (3):(2)$$x_{A}=L\left(1-tan\varphi \right)-\frac{t_{A}}{2cos\varphi }$$
(3)$$y_{A}=0$$

The location of *B*, the midpoint of the upper beam, can be found by determining the coordinates of point *C*, such that (Equation (4)):(4)$$\frac{t_{A}}{\left|OC\right|}=sin\left(2(45^{0}-\varphi \right))=cos2\varphi$$
or
(5)$$\left|OC\right|=\frac{t_{A}}{cos2\varphi }$$
can be written as Equation (5). The coordinates of point *C* are (Equations (6) and (7)):(6)$$x_{C}=L-\left|OC\right|cos\varphi =L-\frac{t_{A}}{cos2\varphi }cos\varphi .$$
(7)$$y_{C}=L-\left|OC\right|sin\varphi =L-\frac{t_{A}}{cos2\varphi }sin\varphi$$

As it can be understood from the geometry, *β* = 2*φ*. The length BC to be used to determine the coordinates of point B from point C (Equation (8)):(8)$$\left|BC\right|=\frac{\left|DC\right|}{2}=\frac{t_{A}}{2cos2\varphi }$$
is indicated as such. The coordinates of the *B* point are obtained by subtracting the horizontal and vertical amounts of the *BC* length from the *x* and *y* coordinates of the *C* point (Equations (9) and (10)).
(9)$$x_{B}=x_{C}-\left|BC\right|sin\varphi =L-\frac{t_{A}}{cos2\varphi }cos\varphi -\frac{t_{A}}{2cos2\varphi }sin\varphi ,$$
(10)$$y_{B}=y_{C}-\left|BC\right|cos\varphi =L-\frac{t_{A}}{cos2\varphi }sin\varphi -\frac{t_{A}}{2cos2\varphi }cos\varphi ,$$

$r_{A/B}$ is obtained by defining point *A* with respect to point *B* (Equation (11)):(11)$$r_{A/B}=\left(x_{A}-x_{B}\right)i+\left(y_{A}-y_{B}\right)j\;$$

*i* and *j* are unit vectors in the *x* and *y* directions. When these points are placed in the equation (Equation (12)):(12)$$r_{A/B}=\left\{L\left(1-tan\varphi \right)-\frac{t_{A}}{2cos\varphi }-\left(L-\frac{t_{A}}{cos2\varphi }cos\varphi -\frac{t_{A}}{2cos2\varphi }sin\varphi \right)\right\}i+\left\{0-\left(L-\frac{t_{A}}{cos2\varphi }sin\varphi -\frac{t_{A}}{2cos2\varphi }cos\varphi \right)\right\}j$$
or Equation (13):(13)$$r_{A/B}=\left\{-Ltan\varphi -t_{2}\left(\frac{1}{2cos\varphi }-\frac{cos\varphi }{cos2\varphi }-\frac{sin\varphi }{2cos2\varphi }\right)\right\}i+\left\{-L+t_{2}\left(\frac{sin\varphi }{cos2\varphi }+\frac{cos\varphi }{2cos2\varphi }\right)\right\}j$$
with a more simplified expression (Equation (14)):(14)$$r_{A/B}=\left\{-Ltan\varphi +t_{2}\left(\frac{\frac{1}{2}\sec\varphi +sin\varphi }{cos2\varphi }\right)\right\}i+\left\{-L+t_{2}\left(\frac{sin\varphi +\frac{1}{2}cos\varphi }{cos2\varphi }\right)\right\}j\;$$
the moment effect of F=−Fi load at point B can be calculated as with Equation (15):(15)$$M_{B}=r_{A/B}\times F=F\left(-L+t_{2}\left(\frac{sin\varphi +\frac{1}{2}cos\varphi }{cos2\varphi }\right)\right)k\;$$

*k* is the unit vector in the z direction. For a dowel beam of unit length in the z direction, the moment of inertia of the corresponding beam is $I=t_{A}{}^{3}/12$ and the distance from the neutral axis to the bottom or top surfaces is $c=t_{A}/2$. The bending stress ($\sigma_{b}$) that will occur in this case can be found using (Equation (16)):(16)$$\sigma_{b}=\frac{6\left|M\right|}{t_{A}{}^{2}}$$

In addition, when the compression (*σ_c_*) and shear (*τ*) components created by the *F* force at the *B* point are considered together, the constraints that can be written to maximize the Poisson effect (*ϑ*) are expressed as follows (Equations (17) and (18)).
(17)$$\sigma_{b}+\sigma_{c}\leq \sigma_{c,all}$$
(18)$$\tau \leq \tau_{all}$$

Another restriction is applied using Equation (19).
(19)$$\sigma_{a}{}^{2}-\sigma_{a}\sigma_{c}+\sigma_{c}{}^{2}\leq \sigma_{c,all}{}^{2}$$

Here, $\sigma_{a}$ and $\sigma_{c}$ are the principal stress values occurring in the natural axes.

The boundary functions described above were used to find the maximum value of the target function ϑ=−tanφ, thus determining the optimum values of *t_A_*, *t_B_* and *φ*.

### 2.3. Numerical Model and Analyses of Auxetic Dowels

In this study, numerical calculations were made using the Abaqus/Explicit v.6.13-1 software (Dassault Systems Simulia Corp., Waltham, MA, USA). Dowel geometry, loading and boundary conditions of the model were based on Figure 7a. In general, an 8-node linear brick, reduced integration, hourglass control element C3D8R was used (about 147,600 elements and 246,800 nodes per model, Figure 7b).

Polylactic Acid (PLA) is modeled as an elastic-isotropic material for numerical analysis. Before analysis, the material properties of PLA were calibrated. As shown in Figure 4a, 0.5 mm displacement was applied from the top surface and pin support was added to the bottom surface of dowel (Figure 7a). At the end of the numeric test, total vertical displacement was 0.5 mm and total horizontal displacement of auxetic pattern was measured. Strain in both directions was calculated using displacements, and Poisson’s ratios of the auxetic patterns were determined with Equation (20).
(20)$$\vartheta =-\frac{\varepsilon_{Trans}}{\varepsilon_{Load}}$$

In Equation (20), $\varepsilon_{Load}$ is the strain in the loading direction; however, $\varepsilon_{Trans}$ is the strain which is transverse to the loading direction.

### 2.4. Production and Experimental Tests of the Dowels

Dowels were designed for production with both 3D printer and injection molding technology. In this study, auxetic dowels were produced with 3D printer (Creality CR-10, made in China) technology with PLA, which is commonly utilized for production in 3D printing technology. Firstly, 3D models of the designed dowels were modeled in the Autodesk Inventor software. Then, based on the CAD models, STP and STL models were prepared for numerical calculations and 3D printing, respectively.

PLA has a glass-transition at about 55 °C and a melt temperature of about 175 °C. Processing temperature needs to be above 190 °C, and the Modulus of Elasticity (MOE) of PLA is 3750 MPa [50]. In this study, nozzle temperature was 220 °C for processing and heat-bed was 70 °C during the dowel production process (Figure 8a).

Uniaxial compression tests were applied to the auxetic dowels in the radial direction for obtaining Poisson’s ratio of the cross-section. Poisson’s ratio tests were carried out on a 5 kN capacity numerically controlled universal testing machine with a 1 mm/min loading rate under the static uniaxial loading (Figure 8b). Loading was continued until the 1 mm deflection occurred in the vertical (Y) direction. In order to obtain Poisson’s ratio of dowels, as seen in Figure 8b, a reference ruler was placed behind the dowels. Two pictures of the dowels were taken, one before loading and the other at the time of 1 mm deformation in the vertical (Y) direction. Then, dowel strains in vertical and horizontal directions were analyzed using the National Instruments IMAQ Vision Builder 6.1 software (National Instruments, Austin, TX, USA). Poisson’s ratios were calculated by applying the edge detection method in the digital image analysis.

## 3. Results and Discussion

### 3.1. Optimization Results for Cross-Section of Auxetic Dowels

After optimization, all dimensions was determined for numerical analysis and experimental tests. Obtained parameter results are given in Table 2, and the optimized quarter part of the cross-section of the auxetic dowel dimensions are shown in Figure 9a,b for the first and second type of dowels, respectively.

### 3.2. Comparison of the Experimental Results, Numerical Analyses and Analytical Calculations

Poisson’s ratios of the designed auxetic dowels were experimentally, numerically and analytically determined. To provide a practical evaluation of how well the Poisson’s ratios obtained by numerical analyses (FEM) and analytical calculations agreed with the observed Poisson’s ratio results from the actual tests, comparisons of the observed test results with the results obtained with FEM and analytical calculations are given in Table 3.

As seen in Table 3, the Poisson’s ratios calculated by numerical, experimental and analytical analyses are consistent with each other. For both dowels, negative Poisson’s ratios were obtained from analytical, numerical and actual tests. It means that these designed, optimized and produced dowels showed auxetic properties. This table also shows the consistency that with the increasing diameter of the dowel, Poisson’s ratios decreases; concordantly to this, the Modulus of Elasticity (MOE) increases. The MOE of the core substance of the first type of dowel was determined as 155.54 MPa, and the MOE of the second type of dowel was determined as 436.76 MPa. All MOE values were calculated with an analytical method and all MOE values were calculated for the core of the dowels. According to Table 3, the difference between the experimental and numerical Poisson’s ratio of the first type of dowel is 13%, while the difference between that of the experimental and analytical is 10%. When the second type of dowel’s Poisson’s ratio results are compared, the difference between the experimental and numerical values is 9%, and the difference between the experimental and analytical results is 18%.

## 4. Conclusions

In this study, it was considered that the property of negative Poisson’s ratio could be used to design different types of dowels which have auxetic cross-sections for easier push-in and harder pull-out for the frame- and panel-type furniture joints. This study was conducted to experimentally, analytically and numerically analyze the Poisson’s ratio of different sizes of auxetic dowels produced from PLA for furniture joints. In the scope of the study, it was planned to design two different types of auxetic dowels with different diameters and sizes for frame- and panel-type furniture joints.

The results of the study provide analytical, numerical and experimental information on the Poisson’s ratio of designed auxetic dowels. According to the results, auxetic behavior was obtained for both sizes of designed dowels. Furthermore, optimum dimensions for cross-sectional geometry of the dowels were calculated and recommended by means of optimization techniques. The designed, optimized and produced dowels could be utilized for one-time ready to assemble (RTA) frame- and panel-type furniture constructions. Therefore, it could be suggested that, for future studies, the auxetic dowels should be tested with furniture joint specimens, and also with 1/1 scaled whole frame- and panel-type furniture.

The auxetic dowels that were designed in this study can be produced with 3D printer technology and an injection molding system as well. The effect of the production technologies on the Poisson’s ratio and strength of the dowels can be investigated in future studies.

## Figures and Tables

**Figure 1 materials-16-02838-f001:**
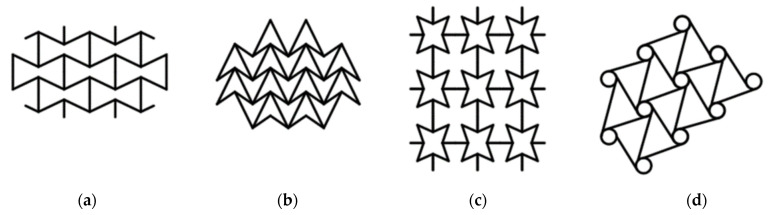
Example auxetic patterns. (**a**) Hexagonal unit cell, (**b**) 3-star, (**c**) 4-star and (**d**) 2D hexachiral [49].

**Figure 2 materials-16-02838-f002:**
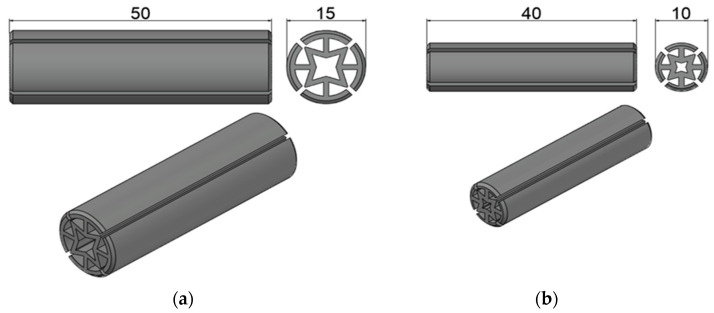
Designed and modeled dowels for frame- (**a**) and panel (**b**)-type (dimensions are in mm).

**Figure 3 materials-16-02838-f003:**
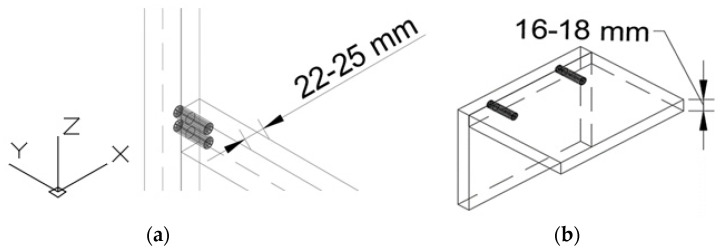
(**a**) Frame joints with the first type of auxetic dowel, (**b**) panels joint with the second type of auxetic dowel.

**Figure 4 materials-16-02838-f004:**
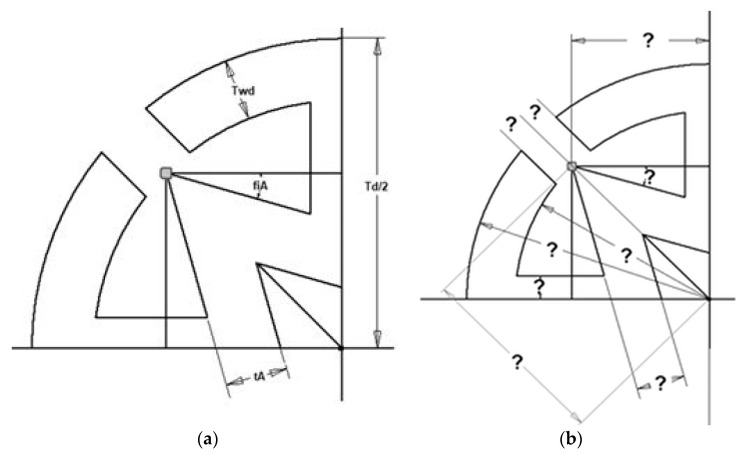
(**a**) Parameters whose determined minimum and maximum level for optimization; (**b**) dimensions calculated by using algorithm for auxetic pattern of dowels (mm).

**Figure 5 materials-16-02838-f005:**
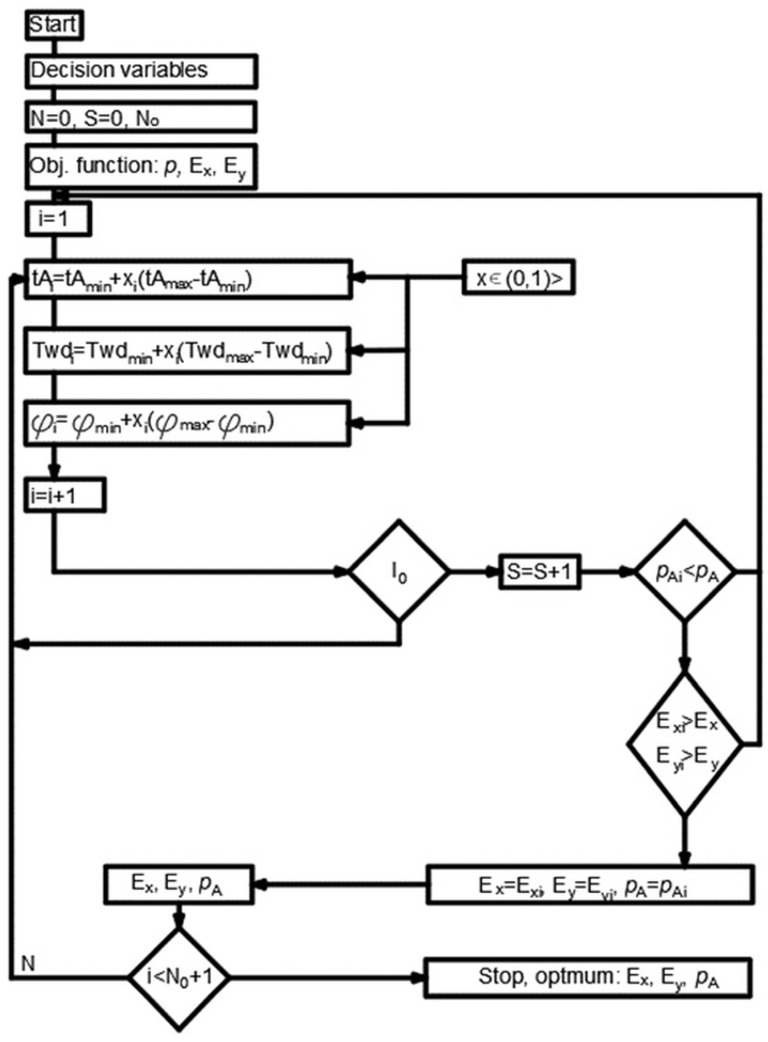
Optimization algorithm of auxetic dowels.

**Figure 6 materials-16-02838-f006:**
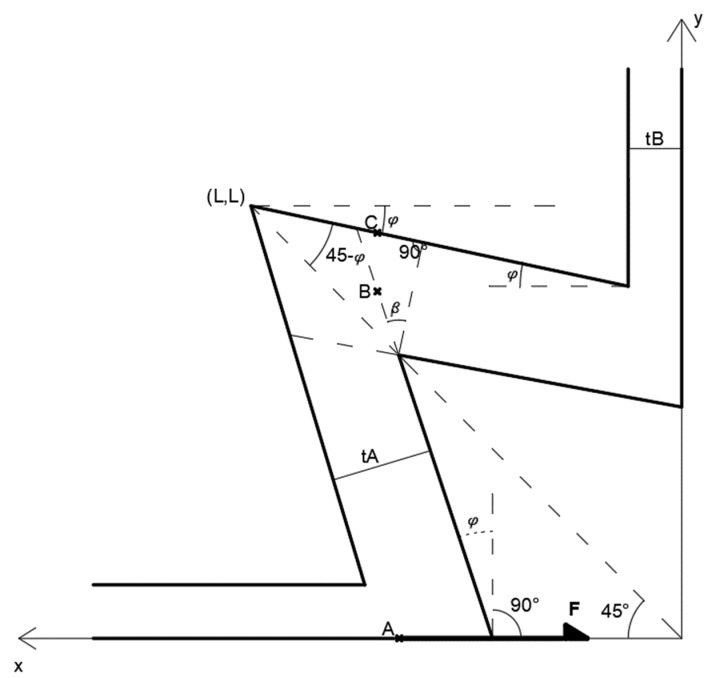
Dowel cross-section and force applied point.

**Figure 7 materials-16-02838-f007:**
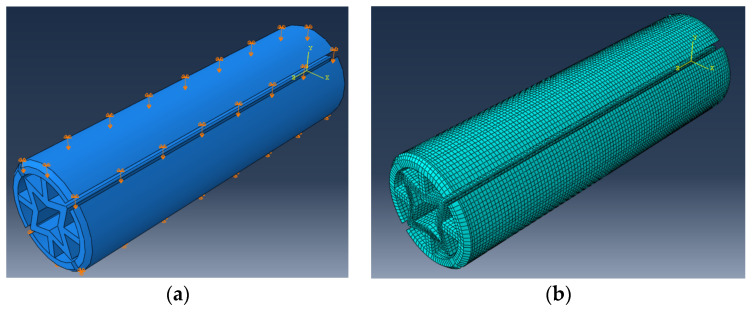
(**a**) Dowel geometry, loading and boundary conditions of the model; (**b**) mesh model.

**Figure 8 materials-16-02838-f008:**
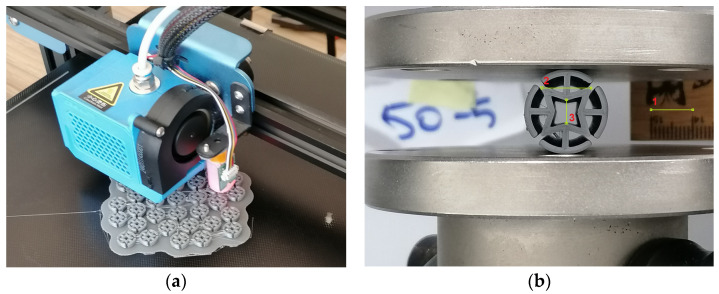
(**a**) Production process with 3D printer technology, (**b**) test setup for determining Poisson’s ratio.

**Figure 9 materials-16-02838-f009:**
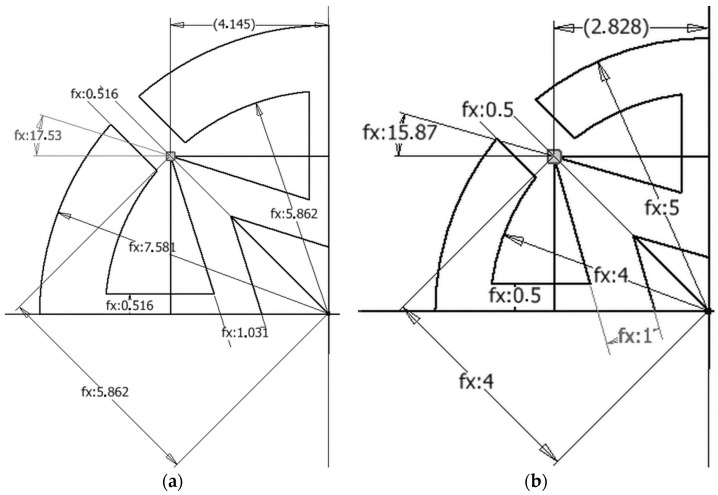
(**a**) First dowel type dimension, (**b**) second dowel type dimension after optimization.

**Table 1 materials-16-02838-t001:** Minimum and maximum levels of dowel dimensions for optimization.

Parameters	1st Dowel Type		2nd Dowel Type	
	Minimum Level	Maximum Level	Minimum Level	Maximum Level
$t_{A}$	1 mm	2 mm	1 mm	2 mm
$\varphi_{A}$	15°	40°	15°	40°
$T_{wd}$	1 mm	2 mm	1 mm	2 mm
$T_{d}$	15 mm		10 mm	
Length	50 mm		40 mm	

**Table 2 materials-16-02838-t002:** Optimized parameters for auxetic dowels.

Parameters	1st Dowel Type	2nd Dowel Type
$t_{A}$	1.03 mm	1 mm
$\varphi_{A}$	17.53°	15.87°
$T_{wd}$	1.72 mm	1 mm
$T_{d}$	15 mm	10 mm
Length	50 mm	40 mm

**Table 3 materials-16-02838-t003:** Poisson’s ratio values of the dowels from numerical, experimental and analytical analyses.

Dowel Type	Dowel Diameter (mm)	Dowel Length (mm)	Poisson’s Ratios			MOE (MPa)
			Experimental	Numerical	Analytical	
1st type	15	50	−0.273 (0.054) *	−0.309	−0.302	155.54
2nd type	10	40	−0.287 (0.065)	−0.313	−0.341	436.76

*: Values in parentheses are standard deviations.

## Data Availability

The data used to support the findings of this study are available from the corresponding author upon request.

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
