# Peer review of "Optimization of the Cross-Sectional Geometry of Auxetic Dowels for Furniture Joints"

_materials, 2023, doi:10.3390/ma16072838_

Round 1

Reviewer 1 Report

The authors tried to optimize the Cross-Sectional Geometry of Auxetic Dow- 2 els for Furniture Joints to have sufficient elastic rigidity and optimum strength using monte Carlo method.

In recent days, the auxetics gain lot of interest in structural application due to their unique properties (better indentation with negative Poisson's ratio). The authors can look into some recent literature and consider them for inclusion. The below literature gives a wide variety of structures for design and key challenges for Realtime implementation.

Kelkar, Parth Uday, et al. "Cellular auxetic structures for mechanical metamaterials: A review." Sensors 20.11 (2020): 3132.

Balan, Madhu, Johnney Mertens, and MVA Raju Bahubalendruni. "Auxetic Mechanical Metamaterials and their Futuristic Developments: A state-of-art Review." Materials Today Communications (2022): 105285.

Ren, Xin, et al. "Auxetic metamaterials and structures: a review." Smart materials and structures 27.2 (2018): 023001.

Please give the sub captions for figure (a) and (b) of 1. Can be seen from above literature.

Please provide the motivation in selection of these structures.

Provide axis system in figure 3

Pleas use appropriate flowchart symbos for figure 4 and improve the quality.

How are the elastic constants measured? Please state the analytical/theoretical discussion on it

Please provide the justification for monte-carlo method., there exist numerous structural design optimization techniques for this purpose. Moreover, some natural patterns also exhibit these properties; Please refer to the literature on “topology optimization in nature.”

Author Response

Response to Reviewer 1 Comments

Point 1: In recent days, the auxetics gain lot of interest in structural application due to their unique properties (better indentation with negative Poisson's ratio). The authors can look into some recent literature and consider them for inclusion. The below literature gives a wide variety of structures for design and key challenges for Realtime implementation.

Kelkar, Parth Uday, et al. "Cellular auxetic structures for mechanical metamaterials: A review." Sensors 20.11 (2020): 3132.

Balan, Madhu, Johnney Mertens, and MVA Raju Bahubalendruni. "Auxetic Mechanical Metamaterials and their Futuristic Developments: A state-of-art Review." Materials Today Communications (2022): 105285.

Ren, Xin, et al. "Auxetic metamaterials and structures: a review." Smart materials and structures 27.2 (2018): 023001.

Response 1: The authors appreciate for the literature suggestions, and also all paper was added to the manuscript.

Point 2: Please give the sub captions for figure (a) and (b) of 1. Can be seen from above literature.

Response 2: Sub captions were added to the figure 1.

Point 3: Please provide the motivation in selection of these structures.

Response 3: Motivation in selection of the structure was added to line 227-229 with sentences below:

“This pattern can be easily applied as honeycomb system and also single, and pattern has auxetic behaviour in both application. Furthermore, 4-star pattern can be produced both with 3D printing system and injection molding system.”

Point 4: Provide axis system in figure 3.

Response 4: Axis system was added to figure 3.

Point 5: Please use appropriate flowchart symbos for figure 4 and improve the quality.

Response 5: Authors would like to thank the reviewer for point out that figure mistakes. Appropriate flowchart symbols were applied.

Point 6: How are the elastic constants measured? Please state the analytical/theoretical discussion on it.

Response 6: All MOE values were calculated with analytical method and all MOE values were cal-culated for the core of the dowels. And this situation has allready explained in the result and discussion section.

Point 7: Please provide the justification for monte-carlo method., there exist numerous structural design optimization techniques for this purpose. Moreover, some natural patterns also exhibit these properties; Please refer to the literature on “topology optimization in nature.”

Response 7: Monte Carlo method is easy and one of the fast method to achieve the optimum results of objective function. Topology optimization is related with decreasing the material volume and find the best distribution of the materials (corss section). Topolgy optimization can be used for the current problem. However, authors concern that this method may effect the cross section of the dowel geometry and it can be difficult to obtain the auxetic behavior. Because of that authors decided to choose Monte Carlo Method in this manuscript.  

Reviewer 2 Report

Dear Athors

This study aimed to optimize the geometry of auxeticc dowels using nonlinear programming, and did tests to validate it. The paper contributes to joints of wood products. However, there are some issues need to be modified. See attachment.

Reviewer

Author Response

Response to Reviewer 2 Comments

Point 1: Indicate each type of auxetic pattern.

Response 1: Figure 1 explanations were written and chosen pattern was corrected.

Point 2: As mentioned, only Poisson's ratio was tested and numerically analyzed, but the load resistance of dowels were not compared and evaluated.

Lowest Poisson's ratio? You used negative value or absolute value?

Response 2: For the dowels, withdrawal strength is the most important strength parameter. However, ın this study, only modulus of elasticity (MOE) values were analytically calculated and compared for each dowel types. Lowest Poisson’s ratio was chosen as it has negative value for obtainig the auxetic behaviour. Withdrawal strength and joint strength of these dowels will be studied in the future works.

Point 3: Figure 5a should be cited here.

Response 3: Figure 5a was cited at line 261, regard to reviewer comment.

Point 4: How did you determine these constraints, which must be declared clearly..

Response 4: Constraints explanation was added to the manuscript.

“Minimum and maximum values of constraints are related with nozzle diameter for 3D printer technology and mold for injection molding systems. Angle of the pattern levels was chosen between 15° and 40° because the length of the arm of the star reached the negative values in the formulas. Because of the reasons above minimum and maxi-mum parameters were determined and given in Table 1.”

Point 5: PLA Define it when first appear..

Response 5: Polylactic Acid (PLA) defined at the first appear place.

Point 6: Indicate the type and enterprise and country of the 3D printer.

Response 6: Enterprice and the country of the 3D printer were added.

Point 7: How many? (Pictures were taken during the tests)

Response 7: An explanation was added to the related to the determining the Poisson’s ratio of the dowels.

“In order to obtain the Poisson’s ratio of dowels; as seen in Figure 8b, a reference ruler was placed behind the dowels. Two pictures of the dowels were taken one before loading and the other at the time of 1 mm deformation in vertical (Y) direction.”

Point 8: According to your description, you prepare to use digital image correlation method to measure the strain, but it is not clear. Describe it in details.

Response 8:  This method is similar to the digital image correlation method. Photos were taken with professional high quality photographic apparatus but the analyses were prepared with another software (Imaq Vision Builder) which is not combined with the photographic apparatus. An explanation was was added to the related to the determining the Poisson’s ratio of the dowels.

“In order to obtain the Poisson’s ratio of dowels; as seen in Figure 8b, a reference ruler was placed behind the dowels. Two pictures of the dowels were taken one before loading and the other at the time of 1 mm deformation in vertical (Y) direction.”

Point 9: This figure (9) is not readable, too ambiguours to read.

Response 9:  Figure 9 was improved as requested.

Point 10: Double check the reference and modify the format to make all references identical.

Response 10: References and abbreviations were checked and corrected.

Round 2

Reviewer 1 Report

The manuscript is well received . No further comments 

All the best